# HopNet: Harmonizing Object Placement Network for Realistic Image Generation via Object Composition

## Abstract

*Realistic image generation is an increasingly desired, but deceptively complicated computer vision task, especially when a specific object is required. Whether generating product advertisements or building novel datasets, object composition for realistic image generation depends on realistic object placements as well as believable object harmonization. To address this task, we introduce HopNet, the first network designed for end-to-end realistic image generation via object composition. HopNet excels in two pivotal tasks: object placement and harmonization, setting state-of-the-art performance in both domains. Unlike conventional methods that employ separate models for each task, HopNet seamlessly integrates object placement and harmonization to acquire knowledge of correlated information. It leverages a transformer-based framework to encode both foreground objects and background scenes and learns attention mechanisms crucial for both object placement and harmonization concurrently. We introduce a modified sparse contrastive loss, allowing our model to learn from multiple both good and bad placements while also learning object harmonization in a self-supervised manner. HopNet generalizes well on challenging scenes while removing the compounding errors associated with using separate models for each subtask.*

## 1. Introduction

Image composition is an increasingly important area drawing growing attention for applications such as image editing, content generation, and the creation of synthetic data for training machine learning models. Traditionally, the process involves a series of manual steps—handling placement, blending, harmonization, and shadowing separately [25, 26]. This segmented approach is not only time-consuming but also hampers performance because each individual task is executed by a separate network. As a result, valuable contextual information is lost between steps, lead-ing to accumulated errors. In contrast, an end-to-end unified model that processes all these tasks together preserves crucial context and interdependencies, resulting in superior overall performance and higher-quality compositions.

Object placement is perhaps the most important step in creating a realistic composite image. A realistic object placement must meet some criteria such as demonstrating correct relative size of the object and believable location. This requires some essential properties such as demonstrating a supporting force (not floating) and being in a semantically reasonable place. Several attempts have been proposed to accomplish this task, with different approaches. These range from: 1. Individual Placement classification [8, 21, 23, 27, 51]. 2. Foreground transformation prediction as a regression problem [1, 15, 18, 19, 22, 33], a reinforcement learning problem [43], and a diffusion problem [2, 22] 3. a Multimodel LLM based common sense approach [43], and 4. Generating 3D placement plausibility heatmaps [52] predicting likelihood scores for all possible location-scale pairs.

Even after selecting the most suitable location, achieving a seamless integration between the foreground object and the background image remains a challenge. Harmonization seeks to address this issue by smoothing the brightness and contrast transitions between the foreground and background images brought about by different lighting conditions. In the context of harmonization, various existing methods approach the problem in diverse ways [6, 11, 12, 35, 37]. For harmonization, both statistical techniques and deep learning methods have been introduced, catering to both low-resolution and high-resolution images [4, 7]. Additionally, the subject of blending becomes relevant when the boundary of the foreground image is not well-cropped [25, 26]. With the incorporation of advanced segmentation models for object cropping [13], this issue occurs less frequently. Lastly shadowing becomes a final step when inserting an object into a background that requires the presence of shadows [25, 26], especially in the case of outdoor scenes.

To perform realistic object composition, one typically needs to navigate through various models encompassing

placement, harmonization, blending, and shadowing, introducing errors and inefficiencies along each step. In this paper, we introduce HopNet, a novel end-to-end transformer-based model that seamlessly combines object placement and harmonization within a single model, maintaining the same model size and latency as previous approaches while improving results from multi-task learning [6, 52]. HopNet shows state-of-the-art performance for both object placement and harmonization in standard metrics. Our approach draws inspiration from prior works [6, 52] but makes several key improvements benefiting from a refined loss function, architecture improvements, and the introduction of multi-task learning.

Our contribution can be summarized as follows:

- We introduce HopNet, a novel transformer-based architecture that can simultaneously handle both object placement and harmonization, while retaining a simple and efficient network architecture.

- We introduce a modified sparse contrastive loss function, enabling the model to learn from scenes with multiple positive and negative ground truth placements

- Self-supervised training paradigm for learning object harmonization.

- Extensive experiments on a large-scale object placement and object harmonization datasets showing state-of-the-art performance.

## 2. Related Work

Realistic image composition is generally performed by first generating a desired foreground object placement and then harmonizing the foreground object to match the background scene. Recently some image generative methods have also performed well. These topics will all be discussed below.

### 2.1. Object Placement

The object placement task consists of predicting viable size and locations for a given foreground object in a specified background scene. This can generally be formulated either as a regression problem or a classification problem.

#### 2.1.1 Object Placement Regression

In regression-based approaches, the goal is to predict a transformation directly—either by estimating a transformation matrix or by generating a bounding box for object placement. This has been approached many ways. Tan et al. [33] predicts bounding boxes for a specific class given a background scene. Similarly, [15] first predicts saliency maps for the object class and then determines specific object

instances that are suitable for that placement. [18] extends the object placement problem by also predicting pose.

GANs are popular architectures for this problem as both the generator and discriminator networks can be used to determine acceptable object placements. In [34] (TERSE) the generated image is discriminated against random other real images. Compositional GAN [1], ST-GAN [19], and SAC-GAN [46] improve by generating and discriminating between placements of the same object. PlaceNet [41] is another adversarially trained model and is the first model that inpaints objects out of a scene to relearn their original position. This is very popular now as a self-supervised learning paradigm.

GracoNet [50] uses a different strategy and views object placement as a graph problem, treating each background image patch as a graph node and learning the best connection for the foreground object node. In IOPRE ( [43]) reinforcement learning is used to learn object transformations. Conditional Transformation Diffusion for Object Placement [22] instead generates object transformations with a diffusion model. Another interesting approach is shown in Think before Placement [29] where a multi-model language model learns to predict a custom regression token and [20] also use a similar process for a 3D scene.

#### 2.1.2 Object Placement Classification

An alternative approach frames object placement as a classification problem, where the task is to determine whether a proposed object placement is realistic. Then every potential location and scale may be evaluated to judge whether the placement is realistic.

The OPA dataset [23] is the first dataset specifically designed for this task. It is a subset of the COCO dataset [21], created by cropping foreground objects and compositing them into new background scenes. These synthetically generated images are evaluated for realism using criteria that include appropriate scaling, the existence of supporting forces, semantic compatibility, minimal occlusion, and consistent perspective, among other factors. Alongside this dataset, the authors introduced SimOPA, a simple ResNet-based classifier [8] trained on the OPA dataset. Building on SimOPA, FOPA (Fast Object Placement Assessment) [27] extends the approach by producing rationality scores for various placement candidates while maintaining similar performance at significantly increased speeds. GALA [51] employs a retrieval network that, trained via contrastive learning on masked backgrounds and corresponding foregrounds, finds the most suitable object given a background and its corresponding placement bounding box. This method can be extended with a sliding-window grid search to generate object placements. Finally, TopNet [52] further refines this strategy by assessing rationality across

all locations and scales, thereby creating a dense heatmap that highlights ideal object placements. As this evaluates all locations and scales at once, it is much faster to sample from than other classification methods.

## 2.2. Object Harmonization

Object harmonization involves adjusting an object so that it seamlessly fits into a different scene, particularly when blending images with varying lighting conditions and environments. For example, one might need to generate a nighttime training dataset using objects originally captured in daylight.

Deep Image Harmonization [35] introduced one of the first end-to-end deep convolutional networks for this task. SSH [11] later proposed a self-supervised framework that leverages a dual data augmentation engine to create training triplets from natural images. DCCF [37] takes a different route by learning human-interpretable neural convolutional filters—such as those for hue, saturation, and value—in an end-to-end manner. In a similar vein, [12] uses regression to adjust preset human filters like contrast and saturation. More recently, PCT-Net [6] has focused on learning a parameter network in a low-dimensional space to apply pixelwise affine color transformations, while [24] extends this work by incorporating adaptive interval color transformations.

Diffusion models have also found application in image harmonization, although their inherent randomness can be a drawback. For instance, [16] employs a diffusion model for color transfer. Meanwhile, DiffHarmony [48] and DiffHarmony++ [49] have addressed this by implementing additional refinement networks aimed at maintaining consistency of the harmonized object.

## 2.3. Image Composition via Generation Methods

Recently, generative methods have demonstrated the capability to yield exceptionally strong results. Both GANs and diffusion models have gained popularity in image generation tasks due to their ability to produce high-quality outputs [9, 30]. In addition, diffusion models have proven effective at conditioning on specific objects to generate images using exemplar inputs [3, 17, 31, 36, 40, 44]. Their power is further evidenced by their capacity to modify both the object and the surrounding context—addressing occlusion issues and adding realistic shadows showing strong scene understanding [38, 39, 47].

Unconstrained Generative Object Compositing (UGOC) [2] stands as the only other end-to-end realistic image composition model in the literature. This method employs diffusion to simultaneously generate an object mask, determine placement, and inpaint the entire scene. However, like all generative diffusion techniques, it comes with inherent drawbacks. These methods tend to be stochastic, invari-

ably introducing changes to the foreground object and, at times, the background scene. Even when using identity-preserving techniques such as IMPRINT [32], minor alterations to the object—such as changes in pose or shape—are still observed, as evidenced in numerous examples provided in the literature. This characteristic renders diffusion-based generative models unsuitable for applications where the object must remain unchanged and faithful, such as in pose classification or product advertising.

## 3. Methodology

### 3.1. Model Design

Our model, HopNet, is proposed as a complete model for realistic image generation via object composting. This is done end-to-end by generating a placement heatmap as well as a harmonized foreground image. In order to create composite images, the harmonized foreground object is composited onto the background scene at locations sampled from the placement heatmap. As shown in Fig. 1, one branch of the network attends to object placement and another branch addresses object harmonization. For this task the network is given a masked foreground object and a background scene where the foreground object should be composited.

To begin, both the background scene and multiscale foreground object (with mask concatenated) are fed through vision transformers. These vision transformers are initialized with weights from pretrained DINOv2-small [28]. The foreground object is encoded at multiple scales before the embeddings are averaged for the next steps in the model. This is shown to improve scale localization in [2]. Next, the class tokens from the foreground path are concatenated with the background features and the background class token is appended to the foreground features. The purpose of this is to add more context when learning the optimal placement and harmonization, as these tasks are very interdependent. A strong object placement may still look unrealistic if the harmonization does not match and visa-versa.

In the placement path (top path in Fig. 1) the background features go through a placement transformer where context from the foreground image is shared via cross-attention blocks. A convolutional upsampling decoder [45] performs progressive upsampling [45] to generate the final output location-scale heatmap. For our implementation the heatmap output is (18,224,224) corresponding to predicted foreground scale, x, and y locations and we use a minimum scale of 0.15. In the bottom path, the foreground features go through a harmonizing transformer with context from the background scene added through cross-attention. These features are also upscaled with a convolutional decoder into a pixel color transformation (PCT) matrix. This PCT matrix is used to apply a color transformation to the original fore-

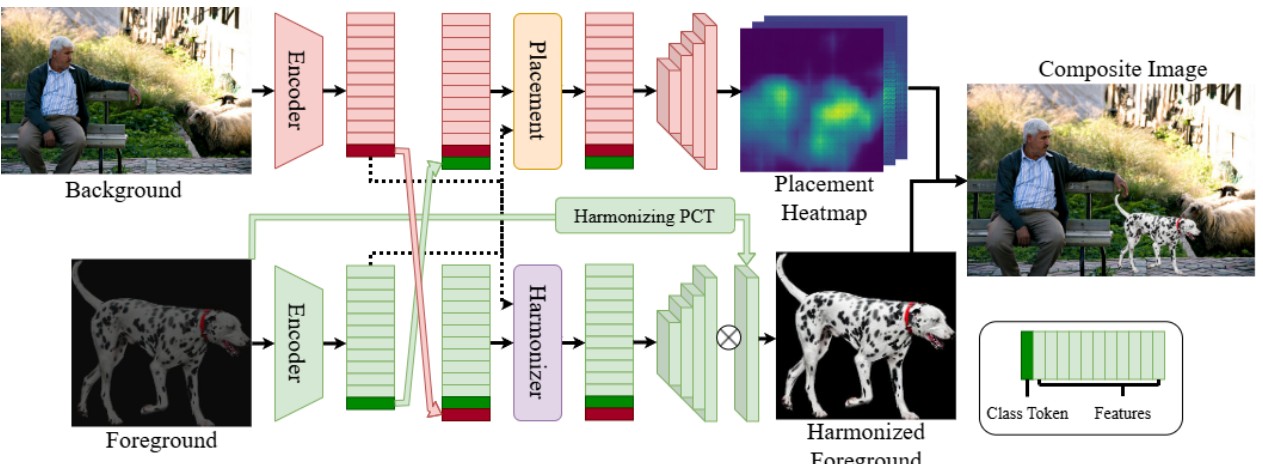

Figure 1. An overview of the proposed HopNet. Foreground and background images are encoded. Class tokens are shared and Harmonizing and Placement transformer blocks learn the respective tasks. Dotted lines are cross-attention. CNN-based upsampling decoders are used in each case.

ground image. This color transformation method ensures that the shape, pose, and orientation of the foreground object is not altered at all aside from the color. The formula for this process is shown in Eq. (1). Here, $p$ refers to the 3 channel RGB pixel values of the original image, $\theta \in \mathbb{R}$ are the upscaled harmonization weights, and $h$ is the final harmonized foreground object. This final foreground object should reflect the original object correctly harmonized to the new background scene.

$$h(p; \theta) = W_\theta \cdot p + b_\theta \qquad (1)$$

$$W_\theta = \begin{pmatrix} \theta_1 & \theta_2 & \theta_3 \\ \theta_2 & \theta_4 & \theta_5 \\ \theta_3 & \theta_5 & \theta_6 \end{pmatrix}, b_\theta = \begin{pmatrix} \theta_7 \\ \theta_8 \\ \theta_9 \end{pmatrix} \qquad (2)$$

### 3.2. Dataset Preparation for Improved Loss and Self-Supervised Training

One improvement that HopNet has over other object placement models is that it is able to learn from multiple positive and negative ground truth locations while other models are only able to learn from one single positive location per training example. HopNet is able to learn from multiple locations by adding all of the known ground truth locations to a margin matrix used in the improved sparse contrastive loss. The OPA dataset [23] is an object placement dataset that provides one ground truth location per training example even though the same foreground object and background scene are used in several labeled ground truth examples. We modify the OPA dataset to instead provide all the positive and negative examples for each foreground-background pair. This provides HopNet with a lot more

context and placement information necessary to produce accurate placement predictions.

The other task that HopNet performs is object harmonization, which is learned in a self-supervised manner. In a ground truth image, both the forground object and background scene will have harmonious lighting. ColorJitter can be applied to both of these to produce an augmented foreground-background pair. In this case the ColorJitter will only affect the brightness, contrast, and saturation of the image, since these are the only parameters that change with different scene lighting. The new augmented pair will also have harmonious lighting with each other, although it will be incongruous with the ground truth lighting. This process gives us two new randomly generated training example pairs in a self-supervised manner (augmented foreground and ground truth background as input with ground truth foreground as target, and ground truth foreground and augmented background as input with augmented foreground as target). Although the HCOCO dataset [5] used for object harmonization evaluation provides augmented foregrounds, this method is used during training to increase training diversity.

### 3.3. Improved Sparse Contrastive Loss and Self-Supervised Harmonization Loss

Object placement is a very interesting problem because it can have many possible correct and incorrect answers even though there are usually very few ground truth locations to learn from. Due to this property, a binary loss function is not the most useful as it only addresses the exact location and scale of the ground truth label, not providing the model very much to learn from. A better approach can be to add a buffer area around the original ground truth location.

This buffer could give labels to many locations immediately around the ground truth location, but it would still ignore or punish other possible good placements in the image.

Instead, we develop a sparse contrastive loss function paired with a range loss for the object placement loss function. The sparse contrastive loss works by encouraging the placements to have high probability in the known ground truth locations and penalizing any placement predictions that are close to or higher than the values of the known ground truth placements. This small penalty makes the model learn the known ground truth locations while also allowing it to assign relatively high value to other unknown locations. This process will work for both positive and negative ground truth locations. First, all of the known ground truth locations are expanded using a buffer matrix $(B(s, x, y))$ in Eqs. (3) and (4). Here $B$ is a tensor with the same shape as the location heatmap (scale, rows, cols) and is set to a buffer value of 0.1 at all locations except for locations immediately near the ground truth location where it is instead set to zero. The size of the margin zone around the ground truth location was 20 pixels in the x and y directions and 2 pixels in the scale dimension. Next, the contrastive loss function penalizes any area given a value too close to, or higher than, the value at the original ground truth location ($\hat{H}_{gt}$ in Eqs. (3) and (4), for positive and negative samples). Here $\min \hat{H}_{gt}$ is the minimum value of the predicted heatmap at the known ground truth locations and $\max \hat{H}_{gt}$ is the same for the max while $H(s, x, y)$ is the actual matrix of predicted heatmap values. This pushes the ground truth location to be the most likely, but allows other locations to score highly as well. The $|\cdot|^{+}$ operation in Eqs. (3) and (4) is equivalent to $max(\cdot, 0)$, although a ReLU activation layer is used in practice. Finally, for object placements the heatmap values should range from zero to one. Eqs. (5) and (6) addresses this, pushing the heatmap ground truth location and heatmap min/max to one and zero. Once again, the placement heatmap dimensions are (18,224,224) corresponding to predicted foreground scale, x, and y location. This new placement loss can be written as the sum of above as $\mathcal{L}_{place} = \mathcal{L}_{con+} + \mathcal{L}_{range+} + \mathcal{L}_{con-} + \mathcal{L}_{range-}$.

$$\mathcal{L}_{con+} = \sum_{(s,x,y)} |(H(s,x,y) - \max \hat{H}_{gt} + B(s,x,y)|^{+} \quad (3)$$

$$\mathcal{L}_{con-} = \sum_{(s,x,y)} |\min \hat{H}_{gt} - H(s,x,y) + B(s,x,y)|^{+} \quad (4)$$

$$\mathcal{L}_{range+} = |min(H)| + |1 - \max \hat{H}_{gt}| \quad (5)$$

$$\mathcal{L}_{range-} = |\min \hat{H}_{gt}| + |1 - max(H)| \quad (6)$$

The harmonization loss used in training is a combination of a masked mean squared error loss, a brightness loss, and a contrast loss between the reconstructed foreground and the target foreground image. A brightness and contrast loss is added because brightness and contrast are the main aspects of an image that need to be adjusted during image harmonization and are implemented as the $L_1$ loss between the target and predicted foregrounds. The foreground mask is used to filter out any background pixels that may have been affected by data augmentation or other processes. The complete loss function is the sum of both the placement loss and harmonization loss as $\mathcal{L}_{total} = \mathcal{L}_{place} + \mathcal{L}_{harmonization}$.

### 3.4. Implementation Details

Both the foreground encoder and the background encoder are Dinov2-small encoders using pretrained weights that are further finetuned during model training. The placement transformer as well as the harmonization transformer both also have traditional transformer blocks with an extra cross-attention layer. During training, random zooming, flipping, and color jitter were used as data augmentation. The input and output sizes are both 224x224 pixels, however the output heatmap has 18 channels. Scale index 0 corresponds to a scale of 0.15 and scale index 17 corresponds to a predicted scale of 1.0. A learning rate of 1E-5 was used along with the adam optimizer and cosine scheduling. The models were trained on GPU servers with 16 or 24 GB GPUs and the best models were selected by selecting the model with the best performance on a validation set (random 10% of the training data). Predicted locations are obtained from the heatmap by normalizing the heatmap between 0 and 1 and taking the $argmax(\hat{H})$. For multiple placement predictions a MaxPool3D layer was used to find local maxes with kernel size equal to the buffer size in the sparse contrastive loss (5,41,41). This selection method increases the diversity of placements.

## 4. Experiment

HopNet is the first model tailored specifically for realistic image generation via object composting. In order to create a realistic image it must perform both object placement and object harmonization well. While each of these tasks has been studied independently, there is no other model, currently, that does both. In lieu of other composited dataset creation models, each subtask of Hopnet will be individually evaluated against other models on that task. The OPA dataset is a common training and benchmarking dataset for the object placement task. The HCOCO dataset is also popular for object harmonization purposes, although we first have to inpaint artificially backgrounds since our model performs both placement and harmonization concurrently. **In all of our evaluations HopNet will be performing full image generation while other models will only perform**

Table 1. Placement bounding box evaluation results on OPA dataset. HopNet has the best overall IoU agreement between the predicted and original ground truth object locations. *Metric taken from paper.

| Model | Top5 mIoU ↑ | Top1 mIoU ↑ | Top5 IoU>0.5 ↑ | Top1 IoU>0.5 ↑ |
|---|---|---|---|---|
| TERSE* | - | 0.108 | - | 0.094 |
| PlaceNet | 0.225 | 0.116 | 0.101 | 0.028 |
| GracoNet | 0.247 | 0.159 | 0.190 | 0.080 |
| TopNet* | 0.241 | 0.197 | 0.160 | 0.116 |
| UGOC* | - | 0.196 | - | **0.314** |
| HopNet | **0.305** | **0.227** | **0.230** | 0.191 |

Table 2. FID for measuring similarity between ground truth and prediction and LPIPS for measuring diversity of the predictions. HopNet has been evaluated using the original foreground in the generated composite image since OPA do not has harmonized foreground. *Metric taken from paper.

| Model | Plausibility FID ↓ | Diversity LPIPS ↑ |
|---|---|---|
| TERSE* [34] | 46.16 | 0 |
| PlaceNet [41] | 36.65 | 0.160 |
| GracoNet [50] | 27.75 | 0.206 |
| TopNet* [52] | - | 0.2758 |
| IOPRE* [43] | 19.62 | 0.262 |
| Diffusion w OBFE* [22] | 20.27 | - |
| CSENet* [29] | **17.51** | 0.137 |
| HopNet | 17.65 | **0.306** |

their corresponding subtask.

## 4.1. Object Placement Evaluation

The object placement task is the most important step when creating an image, and, as stated earlier, is a complicated task because there are many possible location-scale pairs that are realistic. Because of this fact, there are several different methods used to evaluate an object placement model. All of the object placement evaluation metrics are based on a HopNet model trained on the OPA dataset and the evaluation is based of the model's performance on the test split of the OPA dataset. The OPA dataset is a set of synthetic composite images created with different foregrounds and backgrounds from the COCO dataset. It consists of 21,000 positive or realistic object placements and 42,000 negative or unrealistic object placements for a total training set of 64,000 images, It also has a test set of 11,000 examples.

One method to evaluate the quality of placements is to measure the Intersection over Union (IoU) of predicted locations versus the known ground truth location [52]. However, measuring the IoU of just one predicted location is not always the best metric because there are many possible placements that an object can be in any given scene. Instead, a top-k approach can be used where the maximum IoU over the top-k predicted placements is used instead. This approach allows the models to generate a sample of possible locations, and as long as one of the predicted locations intersects with the ground truth placement it will score well. Tab. 1 shows the result of this evaluation. It includes results for both the top-1 and top-5 mean IoU (mIoU) values as well as the percentage of time that one of the bounding boxes has an IoU value greater than 0.5, which can be considered as a 'correct' placement. HopNet performs very well compared to all of the other SOTA models. The only model that has better top-1 IoU¿0.5 is UGOC, but HopNet beat it in mIoU. UGOC [2] also has a disadvantage in that the entire model must be ran again for every new placement,

making it much less efficient.

Two other evaluation metrics for object placement are FID score [14] and LPIPS [42]. The FID score is a measure of similarity between two different datasets and can be understood as a measure of the quality of the generated images. If the predicted composite image dataset is similar to the ground truth test dataset it will have a low FID score which indicates high quality placements. LPIPS, on the other hand, is a measure of generation diversity. It is desirable for the placements to have a high diversity because in any given image there should be many placements that are realistic. The FID score was calculated between images in the OPA test set and top-1 composite images generated from HopNet using the same foreground-background pairs while LPIPS was measured from the top-5 generated composite images. For these metrics the original foreground object was used to prevent extra influence from harmonization. These results are shown in Tab. 2 and once again, show that HopNet has top of the line plausibility (FID) and diversity higher than other SOTA models (LPIPS). Additionally, Fig. 2 shows the quality of the generated images.

## 4.2. Object Harmonization Evaluation

Object harmonization is the other task performed by HopNet in order to generate realistic images. HopNet does learn generally how to harmonize foreground images from the OPA dataset, however we argue OPA dataset is not the right choice for evaluating harmonization. This is because the OPA dataset is made by directly copying and pasting different foreground objects to new backgrounds without the use of any sort of object harmonization. To address this concern, the HCOCO dataset was used instead. HCOCO is a popular benchmark dataset for object harmonization, so it is better suited for to evaluate HopNet's ability to harmonize foreground objects.

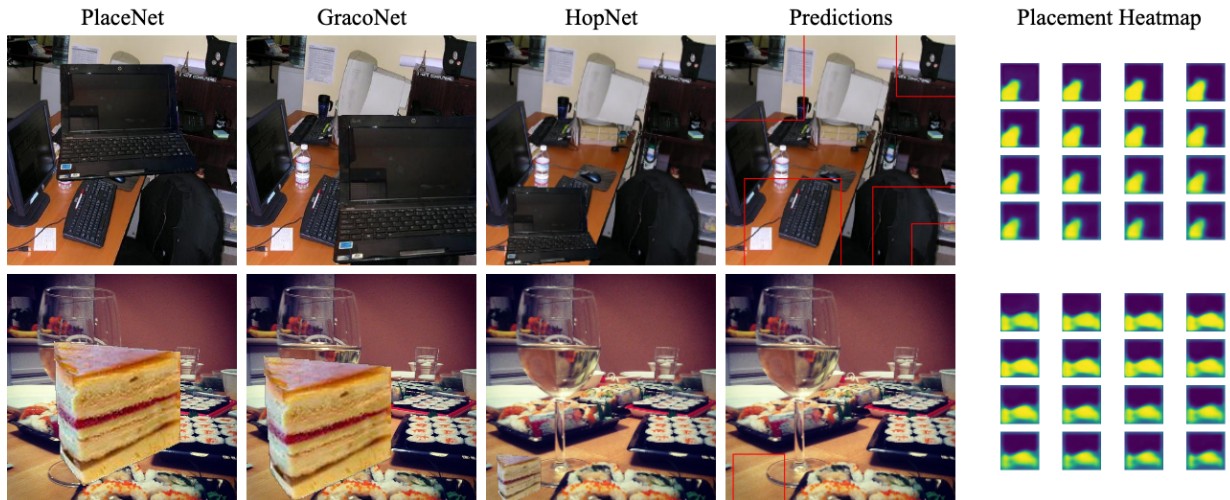

PlaceNet     GracoNet     HopNet     Predictions     Placement Heatmap

Figure 2. A figure showing example composite images from the OPA testing set with HopNet compared to PlaceNet and GracoNet placements (The best models with code and model weights available). The top 5 placement bounding boxes and predicted placement heatmaps are also shown in the last two columns.

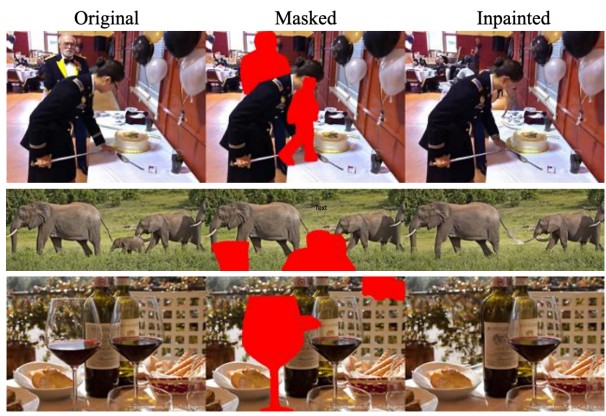

Original     Masked     Inpainted

Figure 3. Inpainting process for the HCOCO dataset. Left is the original image, middle shows the masked areas for inpainting, and right is the inpainted scene with the foreground object removed. Random extra masks were added to reduce the impact of inpainting artifacts.

In the HCOCO dataset individual foregrounds have been segmented and altered into different variations. This dataset provides foreground masks, as well as original and altered images. Overall, there are 60,000 different images, with 25,000 different original images and 35,000 artificial synthetic images. We extend the HCOCO dataset by inpainting the foreground objects out of the original image. This is a common practice in object placement datasets, as it has been used in [41] and [52] for generating training and testing data. For this inpainting task, the pretrained inpainting stable diffusion model was used [10]. It was given the original image, a foreground-combined mask, and the prompt

Table 3. Harmonization evaluation results on HCOCO dataset. HopNet performs very well even with no knowledge of the foreground object location.

| Model | fMSE ↓ | fPSNR ↑ |
|---|---|---|
| Composite images | 1502.99 | 22.72 |
| $Harmonizer$ [12] | 374.96 | 25.16 |
| DCCF* [37] | 317.80 | 25.75 |
| PCT-Net (ViT) [6] | 245.67 | 26.90 |
| DiffHarmony [48] | **153.6** | **29.28** |
| HopNet | 219.83 | 27.40 |

of 'background'. As common practice with other inpainted datasets, extra masks were inpainted to reduce the impact of specific inpainting artifacts. An example of this inpainting process is shown in Fig. 3. This produced realistic and believable backgrounds without the specific foreground present, which is needed as input for the object placement task of our network.

Object harmonization is often evaluated with Mean Squared Error (MSE) and Peak Signal to Noise Ratio (PSNR). However, for both of these signals we are required to use the foreground-specific varients instead (fMSE and fPSNR). This is needed because the foreground-specific metrics are invarient to the foreground location (which may not be the ground truth location with HopNet) and instead shows how well the object is harmonized while disregarding everything except for the foreground. fMSE is simply the mean squared error of the masked foreground and fPSNR is calculated by $fPSNR = 10\log_{10}\left(\frac{MAX_I^2}{fMSE}\right)$ where

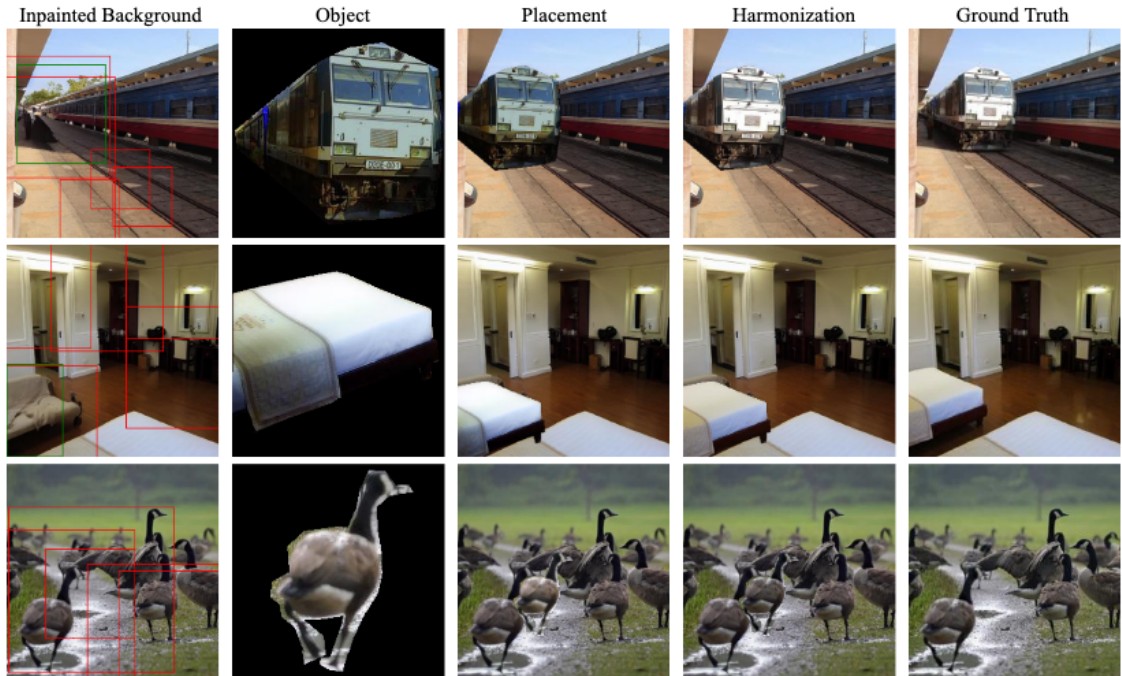

Figure 4. Object placement and harmonization examples on the HCOCO dataset. Both the placements are harmonization results make the generated images appear very realistic.

$MAX_I$ is the maximum pixel value and fMSE is as described above. In Fig. 4 we show both object placement and harmonization examples from the HCOCO dataset. HopNet produces very realistic results, predicting both accurate placements and believable harmonization.

HCOCO evaluation results are compared against other SOTA harmonization models in Tab. 3. From this table it is clear to see that HopNet performs very well, especially when you consider that not only are the other models are focused solely on the task of image harmonization, but they also have full knowledge of where the object is in the scene. Having access to this information is a large benefit because an object in a dark area of a scene will have very different lighting than if it were in a bright region of the same scene. HopNet is not given the location of the object and instead only given the object and inpainted background, so not only is it harmonizing the object to the scene, but it is also predicting where in the scene to place the object. One example of this is shown in Fig. 1 where we can see that the model does not recommend any placements in the very bright light. We show that through cross attention and sharing class tokens HopNet is able to learn from the whole scene and only recommends locations that are realistic for the harmonized foreground it generates. If a specific placement is already known it may be better to use DiffHarmony for object harmonization, but in an end-to-end manner HopNet will choose a plausible location and harmonize well for

that location. More examples of HopNet compositions are shown in the appendix. This figure shows that HopNet creates very believable images, even if the actualy placement or harmonization is different than the original ground truth and is another example of how there are many possible placement and harmonization combinations that are realistic.

## 5. Discussion and Conclusion

HopNet, a novel transformer-based model, pioneers realistic image generation through object compositing. It features two branches: one generating a dense heatmap for foreground placement, and another harmonizing the foreground object with the background scene. Leveraging transformers, both branches encode foreground and scene information, sharing context through class tokens and cross-attention to leverage correlated information.

HopNet outperforms current methods in object placement and harmonization, establishing itself as the state-of-the-art (SOTA) in both tasks and showing the effectiveness of multi-task learning. It offers efficient training and usage, while reducing complexity and compounding errors by replacing several individual models with one unified model. More examples of HopNet image generation can be seen in the appendix.

**Limitations** One limitation of this approach is that it ignores occlusions and shadows in 3D space. This is an activate area of research.

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
