# HopNet: Harmonizing Object Placement Network for Realistic Image Generation via Object Composition

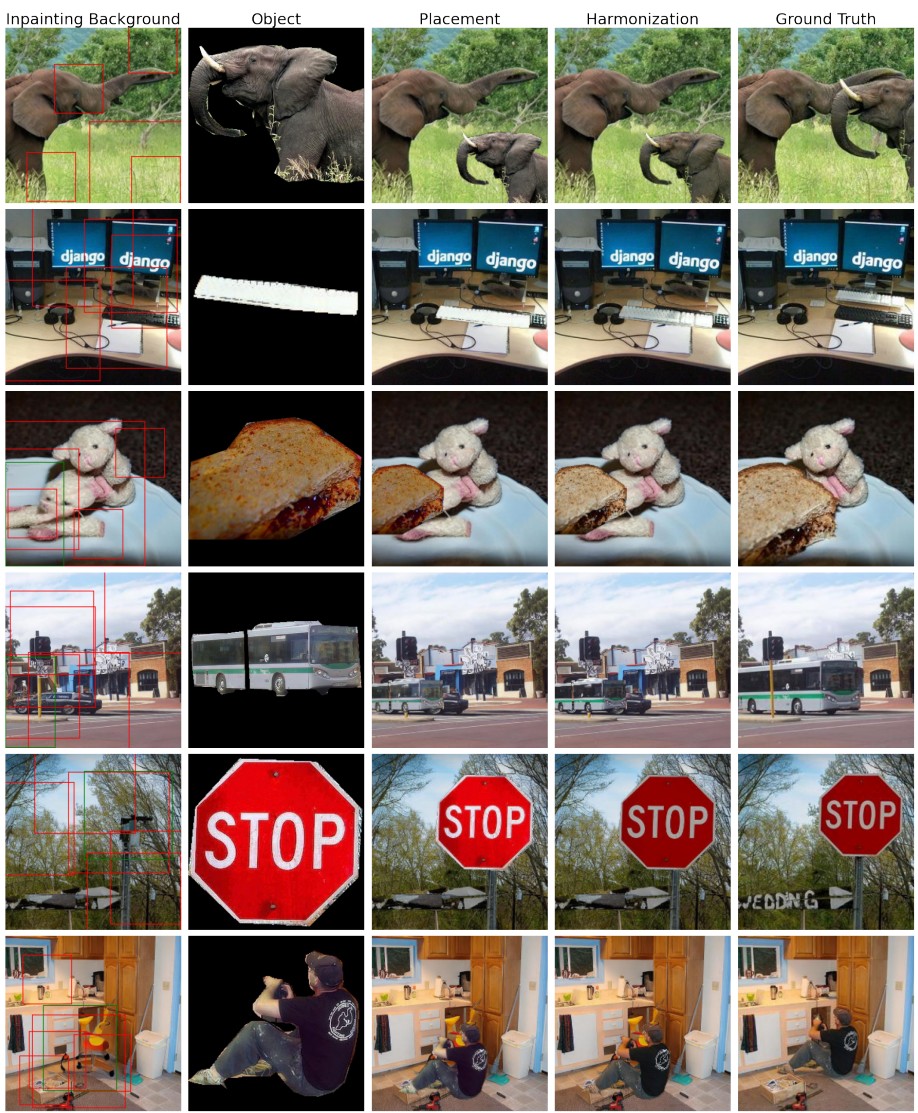

Figure 1. Object placement and harmonization examples on the HCOCO dataset. The first column shows the inpainted background with the top-5 predicted bounding boxes. The second column shows the foreground object. The third column demonstrates the object placement result and the fourth column gives the object placement with harmonization. The last column provides the ground truth images. HopNet's generated images look very realistic, even in examples where the placement is not the same as the ground truth. The objects are also harmonized well so that they match the background scene's lighting.