# OpenReview forum: "HopNet: Harmonizing Object Placement Network for Realistic Image Generation via Object Composition"
_thecvf.com/CVPR/2025/Workshop/CVEU — CVPR 2025_

### Official Review · Reviewer_RmPL · 2025-03-14

**Rating:** 5
**Confidence:** 4

**Review:**

The authors present HopNet, a new framework that can end-to-end learn object placement and harmonization with a dual-branch transformer. The model is trained with contrastive loss so that the model can learn from both positive and negative samples, which is helpful due to the sparsity of the lables in the placement task. The experiments show that the proposed method has improvement over baselines even the baseline model only do single task. The problem is that the proposed contrastive loss is not ablated in the experiment part.

This paper is well-written and the contribution is clear. I suggest to accept this paper into the workshop.

---

### Official Review · Reviewer_KLkG · 2025-03-17

**Rating:** 4
**Confidence:** 4

**Review:**

summary: This paper presents an approach to handle both object placement and  harmonization in a single simple and efficient network architecture, named HopNet. They introduce contrastive loss to learn the representations of good and bad cases in the self-supervised learning paradigm.
pos: 1. sota results in large-scale object placement and harmonization scenarios. 2. the visualizations also show the benefits of the proposed method
neg: some pictures look a little blurred, pls fix this issue.

---

### Official Review · Reviewer_GY1C · 2025-03-23
**Review on HopNet: Harmonizing Object Placement Network for Realistic Image Generation via Object Composition**

**Rating:** 4
**Confidence:** 4

**Review:**

The paper proposes a network for end-to-end image generation by unifying the tasks of object placement and harmonization. It has two branches, one that generates a dense heatmap for foreground placement and another that harmonizes the foreground object with the background scene. It introduces a modified sparse contrastive loss to enable the model to learn from scenes with multiple positive and negative ground-truth placements. Experiments on large-scale object placement and object harmonization datasets demonstrate the method's effectiveness. The paper is well written and the results are solid. The topic fits the workshop very well.

A minor suggestion: Tables 1-3 could be merged into a single two-column table, which better demonstrates the universality of the proposed method on various tasks.

---

### Decision · Program_Chairs · 2025-03-25

**Decision:**

Accept

**Comment:**

The paper introduces HopNet, an end-to-end dual-branch transformer model unifying object placement and harmonization tasks for realistic image generation. Reviewers appreciated its clear contributions, solid empirical results, and the novel use of sparse contrastive loss for handling multiple positive and negative examples. Minor concerns included some unclear visuals and a lack of explicit ablation studies on the proposed contrastive loss.

Given the unanimously positive feedback highlighting its strong performance and clear methodological contributions, the paper is clearly accepted. Authors are encouraged to address reviewers' suggestions, particularly improving visual clarity and providing ablation results on the contrastive loss, in the camera-ready version.